# Hypoxic regulation of chromatin and gene transcription
Jessica D. Kindrick[1,2], Olivia Lombardi[1], Silvia Halim[1], Veronique N. Lafleur[1], Cindy H. Chau [2], William D. Figg [2], Peter J. Ratcliffe [3,4] & David R. Mole [1] ✉

Adaptation to reduced levels of oxygen (hypoxia) is an essential feature of eukaryotic life. Within the animal kingdom, cellular responses are orchestrated by the transcription factor HIF (Hypoxia Inducible Factor) which is regulated by specific 2-oxoglutarate-dependent oxygenases. This family of enzymes also includes histone demethylases, and histone methylation has also been observed to increase in hypoxia. Since histone methylation is associated with gene expression, this has raised questions about whether this also contributes to transcriptional regulation in hypoxia. However, to date pangenomic studies have not been normalised in a way that preserves these bulk changes. Using drosophila chromatin spike-in normalisation, we have shown widespread increases in histone H3K4/9/27/36me3 in hypoxia at almost all gene loci that occur irrespective of whether gene expression is increased or reduced. However, methylation of H3K4me3 and H3K36me3 increases most at direct transcriptional targets of HIF and this is abrogated by inactivation of HIF. Taken together this suggests that global H3 trimethylation increases in hypoxia are widespread and not sufficient to predict transcriptional direction, whereas enhanced H3K4me3/H3K36me3 at direct HIF targets appears consequent to HIF binding and transcriptional engagement.

Oxygen is essential for all eukaryotic life and cells have therefore developed mechanisms for sensing and responding to reduced availability (hypoxia). In multicellular animals, transcriptional responses are largely orchestrated by a family of transcription factors called hypoxia-inducible factors (HIFs). These are regulated through post-translational modifications catalysed by oxygen-sensing members of the 2-oxoglutarate-dependent dioxygenase enzyme family, specifically egl-9 family hypoxia inducible factor 1-3 (EGLN1-3) and factor inhibiting HIF (FIH). The EGLN enzymes all have a high $K_m$ for oxygen such that their activity varies within physiological ranges of oxygen, thereby allowing them to operate as functional oxygen sensors. In addition, several other members of this large family of enzymes have a similar oxygen sensitivity to the EGLNs and FIH and therefore also have the potential to act as oxygen sensors[1,2].

Notable among these are members of the lysine demethylase (KDM) Jumonji C (JmjC) domain enzymes KDM6A, KDM5A and the KDM4s that catalyse histone demethylation. KDM6A, also known as UTX (Ubiquitously transcribed tetratricopeptide repeat, X chromosome) catalyses demethylation of H3K27me2/3, while KDM5A catalyses demethylation of H3K4me2/3 and KDM4 leads to demethylation of H3K9me2/3 and H3K36me2[3,4]. While hypoxia reduces the activity of these enzymes, some histone

demethylases are transcriptional targets of HIF and consequently their abundance increases in hypoxia, potentially offsetting their decreased activity[2].

The presence of these competing effects on histone demethylases has led many groups to examine the effects of hypoxia on global histone methylation. Overall, methylation increases in hypoxia, indicating that the predominant effect arises from inhibition of demethylase activity. Specifically, bulk increases in histone marks associated with both highly transcribed genes, H3K4me3[5–11], and H3K36me3[5,6,12–14], as well as inactive genes, H3K9me3[6,8,12,14–21] and H3K27me3[6–9,12,20,21] have been consistently reported across a range of cell lines and degrees and durations of hypoxia. These reports have largely employed immunoblotting, mass-spectrometry, immunohistochemistry and enzyme-linked immunosorbent assays to examine histone methylation and have not reported locus-specific changes. Nevertheless, in order for bulk changes to be detectable, it is highly likely that these changes in histone trimethylation are widespread across the genome.

To date there have been only limited attempts to examine pangenomic changes in histone methylation in hypoxia and to systematically relate them to changes in gene expression[6,20,22–24]. Furthermore, these studies lack normalisation controls, and the use of conventional normalisation strategies

[1]NDM Research Building, University of Oxford, Old Road Campus, Oxford, UK. [2]Molecular Pharmacology Section, Genitourinary Malignancies Branch, Center for Cancer Research, National Cancer Institute, National Institutes of Health, Bethesda, MD, USA. [3]Ludwig Institute for Cancer Research, University of Oxford, Old Road Campus, Oxford, UK. [4]The Francis Crick Institute, London, UK. ✉e-mail: david.mole@ndm.ox.ac.uk

will obscure the global changes described above, thereby making it difficult to relate changes in histone trimethylation at individual loci to changes in gene expression. Here we undertake pangenomic analyses of histone H3K4me3, H3K36me3, H3K9me3 and H3K27me3 histone trimethylation in normoxic and hypoxic PC3 and HCT116 cells, using ChIP-seq normalised to Drosophila spike-in controls to observe the hypoxic induction seen in bulk analyses. We relate these changes to changes in gene expression using RNA-seq analysis of transcript abundance, normalised to ERCC spike-in control. By relating our findings to ChIP-seq analysis of HIF transcription factor binding and repeating experiments in HIF-1β (also known as ARNT - aryl hydrocarbon nuclear translocator) KO cells lacking a HIF-response we identify HIF-independent global hypoxic increases in histone trimethylation that occur both at upregulated and downregulated genes as well as HIF-dependent increases in H3K4me3 and H3K36me3 occurring in cis at HIF target genes.

## Results

### Hypoxia induces bulk changes in histone trimethylation

We first determined the effect of hypoxia on histone trimethylation in six commonly used cancer cell lines A549 (lung carcinoma), HCT116 (colorectal carcinoma), HepG2 (hepatocellular carcinoma), PC3 (prostate carcinoma), RCC4 + VHL (clear cell renal cell carcinoma stably re-expressing wild type von Hippel Lindau gene), and T47D (infiltrating ductal breast carcinoma). Cells were incubated in normoxia (21% $O_2$) or hypoxia (0.5% $O_2$) for 16 h prior to harvest. Induction of the hypoxia-inducible factors (HIF-1α and HIF-2α) was observed in all 6 cell lines together with induction of the canonical HIF-target gene, NDRG1 (N-myc down-regulated gene 1— not expressed in T47D cells) (Fig. 1). Consistent with previous reports[6,20], induction in bulk levels of H3K4me3 and H3K36me3 was also observed after 16 h hypoxia.

We next examined bulk levels of histone trimethylation using chromatin immunoprecipitation (ChIP). Firstly, titration experiments were performed in which the quantity of antibody and beads was kept constant, whilst the amount of input chromatin was varied (Fig. 2A–D). For each histone trimethylation, the amount of immunoprecipitated chromatin captured varied linearly with the amount of input chromatin. This indicates that the quantity of antibody and beads used are in excess of the amount of chromatin available to be captured. By fixing the ratio of input chromatin to antibody/beads in the middle of this range, we were able to ensure a linear response in subsequent assays. PC3 and HCT116 cells were then incubated in normoxia (21% $O_2$) or hypoxia (0.5% $O_2$) for 16 h prior to chromatin immunoprecipitation. Multivariate analysis by ANOVA of the amount of captured chromatin immunoprecipitated using H3K4me3, H3K36me3, H3K9me3 and H3K27me3 antibodies from both PC3 and HCT116 cells revealed significant induction of histone trimethylation in hypoxia, independent of the specific mark and of cell type (Figs. 2E, F).

In order to determine locus-specific changes in histone trimethylation, the immunoprecipitated chromatin was then analysed by high-throughput sequencing. However, conventional ChIP-seq entails various normalisation steps during library preparation and bioinformatic analysis that obscure global differences in signal between two conditions. Consequently, several previous publications that have shown global induction of histone trimethylation in hypoxia by immunoblotting, were unable to recapitulate the result by ChIP-sequencing[6,20]. We therefore employed an adapted ChIP-seq workflow incorporating ChIP with reference exogenous genome (ChIP-Rx)[25–27] using a Drosophila chromatin spike-in normalisation control that preserved these bulk changes (Fig. 2G). When normalised to Drosophila spike-in control, ChIP-seq analysis of bulk H3K4me3, H3K36me3, H3K9me3 and H3K27me3 trimethylation in PC3 and HCT116 cells again showed significant, independent hypoxic induction by ANOVA comparable to that seen by Qubit quantitation of immunoprecipitated chromatin (Fig. 2H, I). This highlights the importance of including a spike-in control for accurate detection of global epigenetic changes by ChIP-sequencing (ChIP-seq).

## Hypoxia induces histone trimethylation across the genome

ChIP-seq reads were then mapped to individual gene loci. As previously described[28], in the basal normoxic state, H3K4me3 signal was strongest at chromatin flanking transcriptional start sites of actively transcribed genes (Supplementary Fig. 1) in both PC3 and HCT116 cells[28]. Similarly, normoxic H3K36me3 signal mapped to gene bodies of actively transcribed genes. Conversely, the signals for normoxic H3K9me3 and H3K27me3,

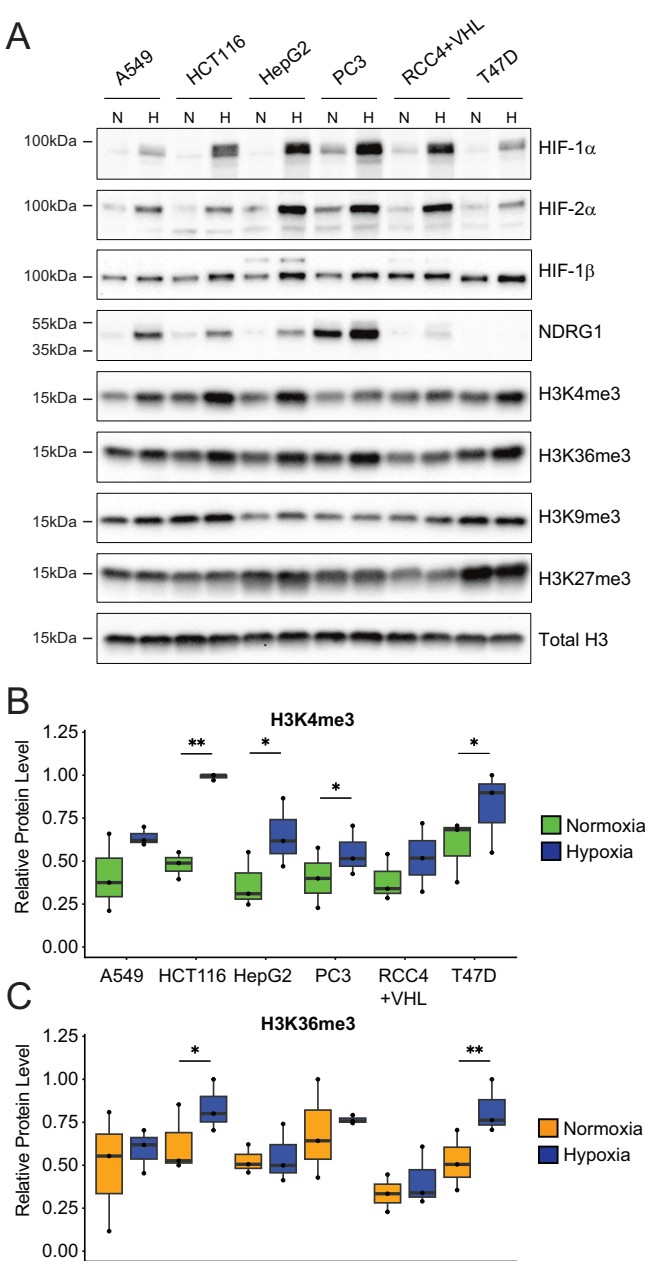

**Fig. 1 | Hypoxia induces both 'active' and 'repressive' histone trimethylation in commonly used cancer cell lines. A** Representative immunoblots of the indicated proteins in whole cell lysates of six commonly used cancer cell lines incubated in 21% $O_2$ normoxia (N) or 0.5% $O_2$ hypoxia (H) for 16 h. Densitometric analysis of **B** H3K4me3 and **C** H3K36me3 levels in normoxia or 16 h of 0.5% $O_2$ hypoxia (depicted in blue) normalised to Total histone H3 signal and averaged across $n = 3$ replicates with individual data points superimposed. Error bars represent the full range of values and significance was calculated by paired Student's $t$ test; *$p < 0.05$, **$p < 0.01$.

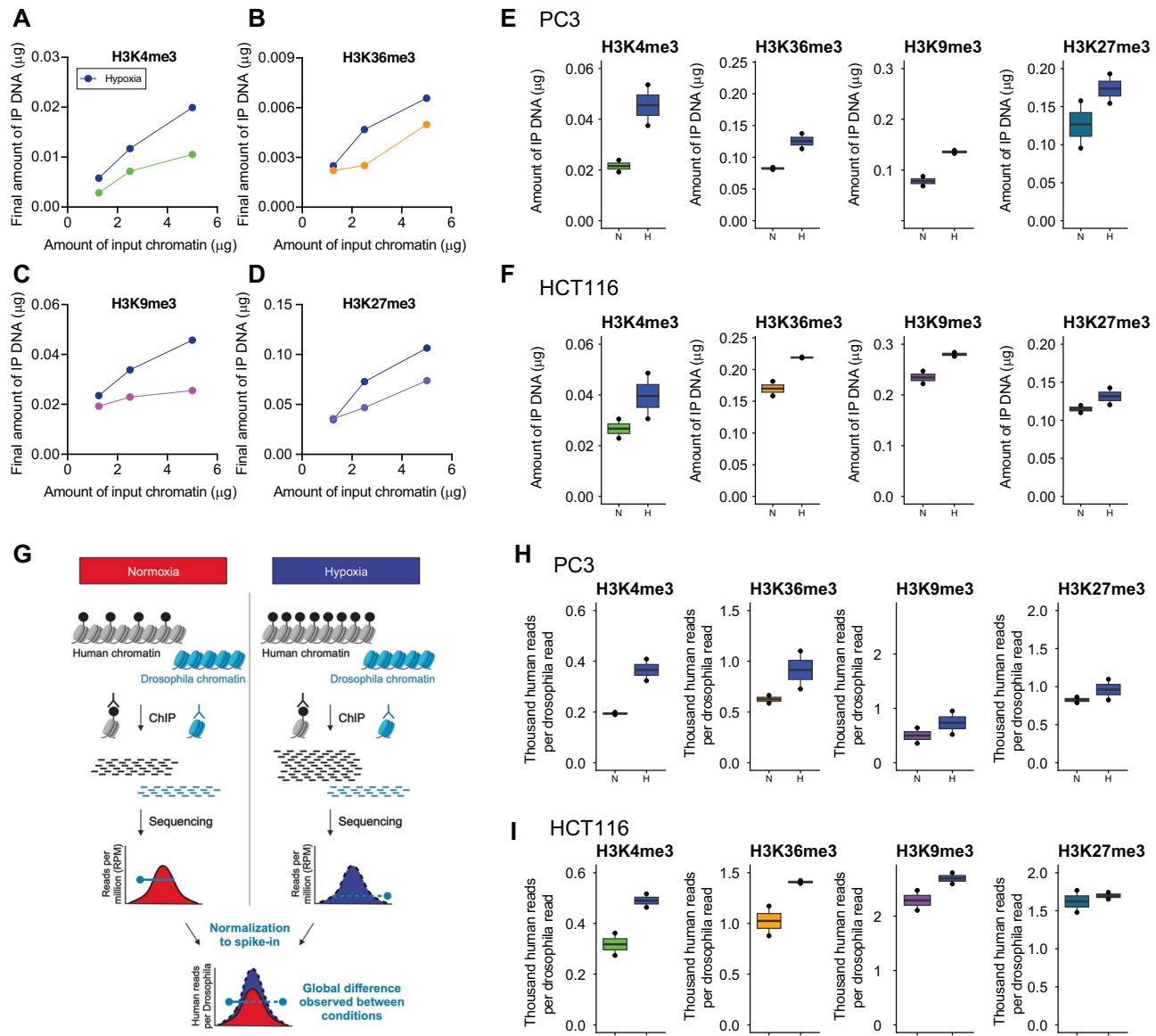

**Fig. 2 | Quantitative ChIP analysis of histone methylation.** Varying quantities of input chromatin (1.25, 2.5 and 5 μg), harvested from PC3 cells incubated in 21% $O_2$ normoxia or 0.5% $O_2$ hypoxia (blue) for 16 h were immunoprecipitated with a constant amount of antibody directed against **A** H3K4me3, **B** H3K36me3, **C** H3K9me3, or **D** H3K27me3. The amount of immunoprecipitated DNA was quantified by Qubit. Qubit quantitation of DNA immunoprecipitated (IP) from **E** PC3 and **F** HCT116 cells incubated in 21% $O_2$ normoxia (N) or 0.5% $O_2$ hypoxia (H, blue) for 16 h, using antibodies directed against H3K4me3, H3K36me3, H3K9me3, or H3K27me3. $n = 2$ independent ChIP experiments. Error bars represent standard error of the mean. Multivariate analysis by ANOVA indicated that the amount of IP'd DNA varied independently with hypoxia ($p = 1 \times 10^{-5}$), cell type ($p = 1 \times 10^{-7}$) and histone mark ($p = 4 \times 10^{-11}$) **G** ChIP-seq normalisation workflow for studying changes in histone modification using Drosophila chromatin spike-in (Created in BioRender. Fontana, T. 2024, https://BioRender.com/a231248). A fixed amount of Drosophila chromatin (pale blue) is added to both normoxic (red) and hypoxic (dark blue) human samples. Chromatin immunoprecipitation (ChIP) for the histone modification recovers more human chromatin (grey) from the hypoxic samples. However, this difference is obscured when signal is normalised to the sequencing depth for each sample. Once each sample is normalised to the internal spike-in Drosophila control, the difference in captured human chromatin between samples is restored. Total read count, normalised to Drosophila spike-in, for ChIP-seq analysis of H3K4me3, H3K36me3, H3K9me3, and H3K27me3 in **H** PC3 and **I** HCT116 cells incubated in normoxia (N) or 16 h at 0.5% $O_2$ hypoxia (H, blue). Multivariate analysis by ANOVA indicated that the Total read count, normalised to Drosophila spike-in varied independently with hypoxia ($p = 1 \times 10^{-3}$, $F = 16.1$, df = 1), cell type ($p = 3 \times 10^{-10}$, $F = 190$, df=1) and histone mark ($p = 8 \times 10^{-10}$, $F = 79.9$, df=3). Error bars represent the full range of values. $n = 2$ independent experiments.

were much more diffuse, but were enriched across the gene bodies of low- and non-expressed genes.

We then compared ChIP-seq signal for each histone trimethylation in normoxia and hypoxia (Supplementary Fig. 2A–H). ChIP-seq profiles averaged across gene bodies and flanking regions in normoxia and hypoxia show that the average signal for H3K4me3, H3K36me3, H3K9me3 and H3K36me3 increase across gene loci in hypoxia in both PC3 and HCT116 cells (Fig. 3A–H).

We then tested whether global changes in total histone H3 occupancy might underlie these changes in histone methylation. Total histone H3 ChIP-seq signal increased slightly in hypoxia, consistent with previously reported increases in overall nuclear compaction[29] (Supplementary Fig. 3). Hypoxic increases in global H3K4me3, H3K36me3 and H3K9me3 signal were still observed, even when normalised to this increased total H3 histone signal. Conversely, global changes in H3K27me3, which have previously been associated with chromatin

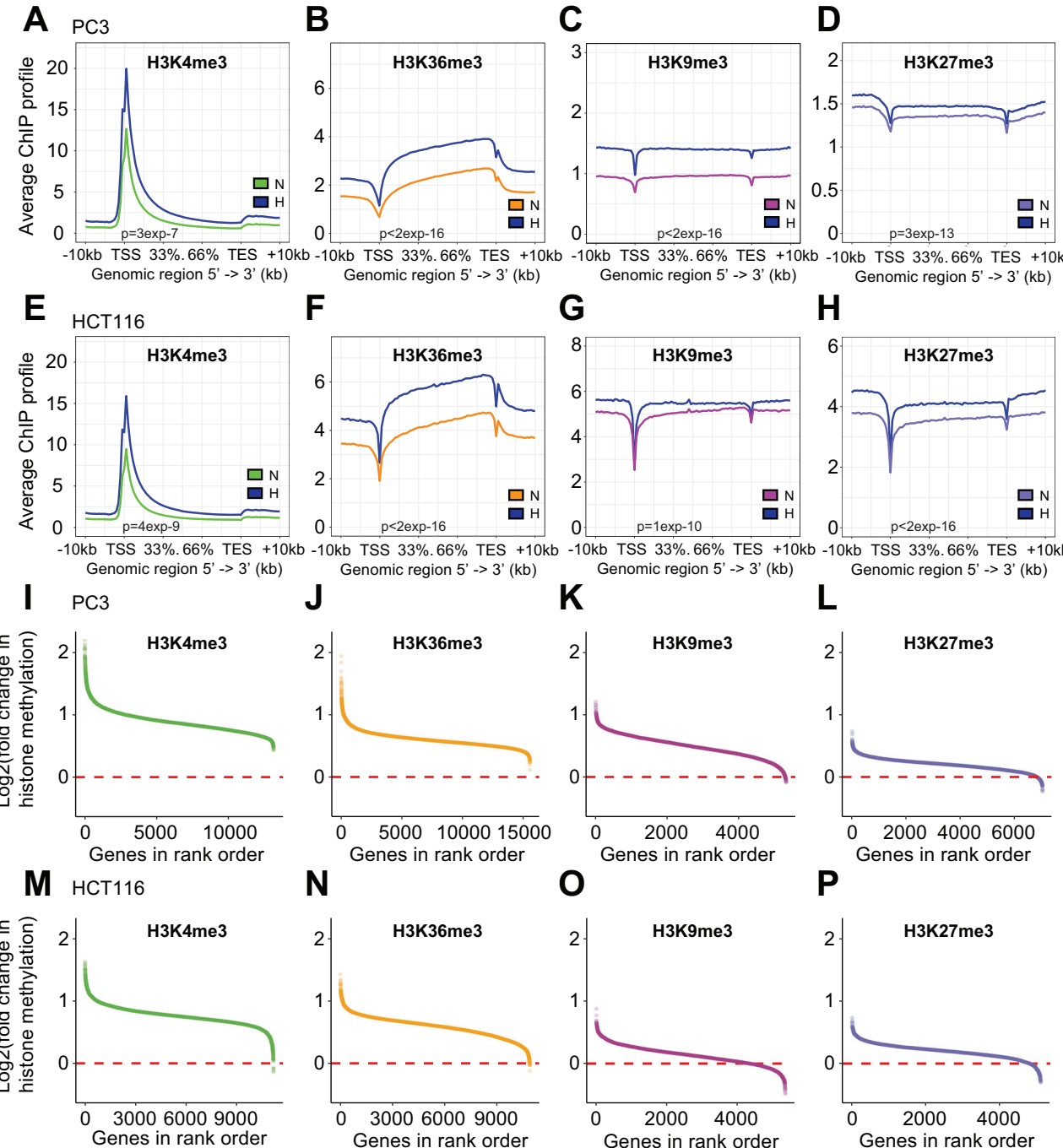

**Fig. 3 | Hypoxia induces global changes in histone tri-methylation across gene bodies.** Average profiles of H3K4me3 and H3K36me3, H3K9me3 and H3K27me3 ChIP-seq signal in **A–D** PC3 and **E–H** HCT116 cells incubated in 21% $O_2$ normoxia or 0.5% $O_2$ hypoxia (blue) for 16 h. Profiles were averaged between $n = 2$ replicates, normalised to Drosophila spike-in control, and plotted across all gene bodies ±10 kb. Multivariate analysis of variance indicated statistically significant differences between signal across gene bodies in normoxia and hypoxia. **I–P** Average ($n = 2$)

log2 fold-change in H3K4me3 and H3K36me3, H3K9me3 and H3K27me3 ChIP-seq signal for each gene was calculated using the total number of ChIP-seq reads across each gene body ±2 kb normalised to Drosophila spike-in control for each condition. The log2 fold-change in the indicated mark between hypoxia (0.5% $O_2$, 16 h) and normoxia (21% $O_2$) for **C** PC3 and **D** HCT116 cells for each gene plotted against its ordered rank.

compaction, were unaltered by hypoxia when normalised to total histone signal.

However, average global changes across the genome do not fully represent the distribution of changes across each gene locus. Therefore, to display the range of responses at different gene loci, we plotted the fold-change in the absolute level of each histone trimethylation at each gene locus against its rank order (Fig. 3I–P). Displaying the results in this way revealed

that the active histone marks, H3K4me3 and H3K36me3, were induced by hypoxia across all gene loci in PC3 cells and almost all gene loci in HCT116 cells (H3K4me3: 11195/11200; H3K36me3: 10888/10895) at which the mark was detected. Similarly, the repressive marks, H3K9me3 and H3K27me3 were also induced by hypoxia across the large majority of loci in both cell lines (PC3 H3K9me3: 5290/5319; PC3 H3K27me3: 6784/7018; HCT116 H3K9me3: 4336/5355; HCT116 H3K27me3: 4794/5103) at which

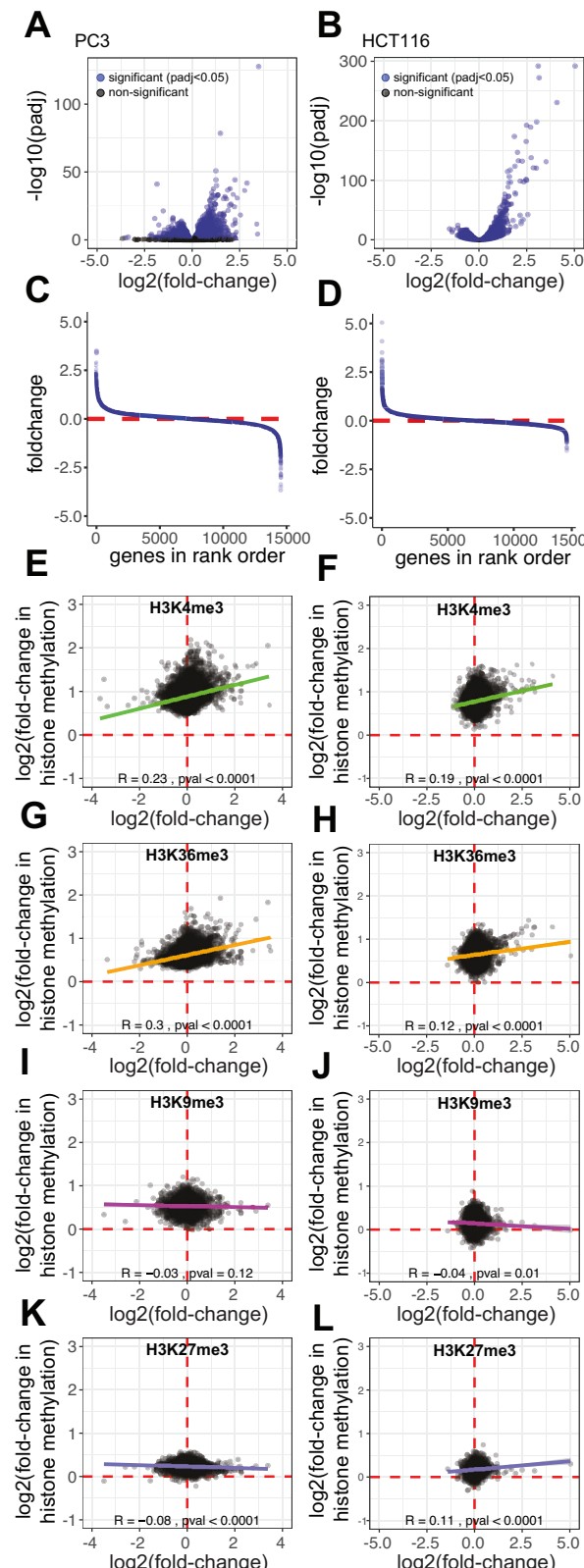

**Fig. 4 | Correlation of hypoxic changes in histone tri-methylation with transcriptional regulation.** Volcano plots showing log2 fold-change in transcript abundance in hypoxia versus -log10 p-value for RNA-seq analysis of **A** PC3 ($n = 4$ replicates) and **B** HCT116 cells ($n = 3$ replicates) incubated in normoxia (21% $O_2$) or hypoxia (0.5% $O_2$, 16 h) following normalisation of each sample to ERCC spike-in control. The log2 fold-change in the transcript abundance between hypoxia (0.5% $O_2$, 16 h) and normoxia (21% $O_2$) for each gene is plotted against its ordered rank for **C** PC3 and **D** HCT116 cells. **E–L** Scatter plots showing log2 fold-change in transcript abundance against log2 fold-change in the indicated histone trimethylation in hypoxia (0.5% $O_2$, 16 h) across each gene body (±2 kb) for PC3 cells and HCT116 cells. Pearson's correlation test was used to determine the correlation between.

## Hypoxia induces histone trimethylation irrespective of gene regulation

Polyadenylated RNA-seq analysis was performed on PC3 and HCT116 cells incubated in normoxia and 0.5% hypoxia for 16 h. Similar to the ChIP-seq approach, an exogenous spike-in standard designed by the External RNA Controls Consortium (ERCC) was used to control for any bulk changes in mRNA abundance (Supplementary Fig. 2I, J). However, in contrast to histone trimethylation, no significant change in total mRNA abundance was observed between cells incubated in normoxia or 0.5% hypoxia for 16 h, with almost identical numbers of genes either up- or downregulated in hypoxia (7,507 up vs 7,025 down for PC3, and 6,793 up vs 7,218 down for HCT116 cells) (Fig. 4A–D).

Hypoxia-induced changes in each histone trimethylation were then plotted against changes in gene expression in PC3 cells (Fig. 4E, G, I, K) and HCT116 cells (Fig. 4F, H, J, L). Notably, both upregulated and downregulated genes showed increases in all four histone trimethylations in both cell lines, as evidenced by log2FC in histone methylation being greater than 0 for both up and downregulated genes (Fig. 4E–L, Supplementary Fig. 4 for average methylation traces at up/downregulated genes and Supplementary Fig. 5 for representative genes). Thus, while basal levels of histone trimethylation correlate well with basal transcript levels, hypoxic changes in histone trimethylation correlate less well with hypoxic changes in histone trimethylation showing similar directional changes irrespective of the change in transcript level.

Nevertheless, weak correlations were observed between hypoxic gene regulation and changes in histone trimethylation. Specifically, increases in H3K4me3 and H3K36me3 positively correlated with hypoxic gene induction in both PC3 and HCT116 cells (Fig. 4E–H). Conversely, H3K9 showed minimal or no correlation with gene expression, whilst H3K27me3 correlated minimally but inconsistently in the two cell lines (Fig. 4I–L).

## HIF target genes are associated with exaggerated hypoxic induction of histone H3K4me3 and H3K36me3 trimethylation

To explore this further in PC3 and HCT116 cells, we made use of previously published ChIP-seq analysis of HIF-1α, HIF-2α and HIF-1β binding in these cells[30,31]. HIF is the primary transcription factor, operating as a transcriptional activator to regulate the hypoxic response. It comprises a heterodimer of an alpha- (HIF-1α or HIF-2α) and a beta-subunit (HIF-1β). HIF target genes were defined as hypoxia-inducible genes adjacent to a canonical HIF-binding site (i.e. both HIF-1α and HIF-1β or HIF-2α and HIF-1β). The criteria used to define HIF target genes were relatively strict. While this should result in a low rate of false positivity, it is likely that a number of HIF target genes will be falsely assigned as non-HIF target genes. HIF target genes exhibited both greater hypoxic induction in H3K4me3 and H3K36me3 levels and a stronger correlation with hypoxic changes in gene expression than non-HIF target genes (Fig. 5 and Supplementary Fig. 6 for signal representative gene loci). Both HIF and activated RNA polymerase 2 (RNApol2) are known to recruit histone methyltransferases to gene loci and HIF may therefore either directly or indirectly increase histone methylation in cis at HIF-target gene loci. We therefore performed ChIP-seq for total RNApol2 (NTD- N-terminal domain), phosphor-Ser5 RNApol2 and phosphor-Ser2 RNApol2 on PC3 and HCT116 cells incubated in normoxia and 0.5% hypoxia for 16 h (Fig. 6). Enhanced, hypoxic induction of total

the mark was detected. However, for all four histone trimethylation marks, there was a wide range in the magnitude of induction by hypoxia between different gene loci, suggesting that they may correlate with differential effects on gene expression between loci. We therefore sought to test whether this variation in histone trimethylation correlated with hypoxic changes in gene transcription.

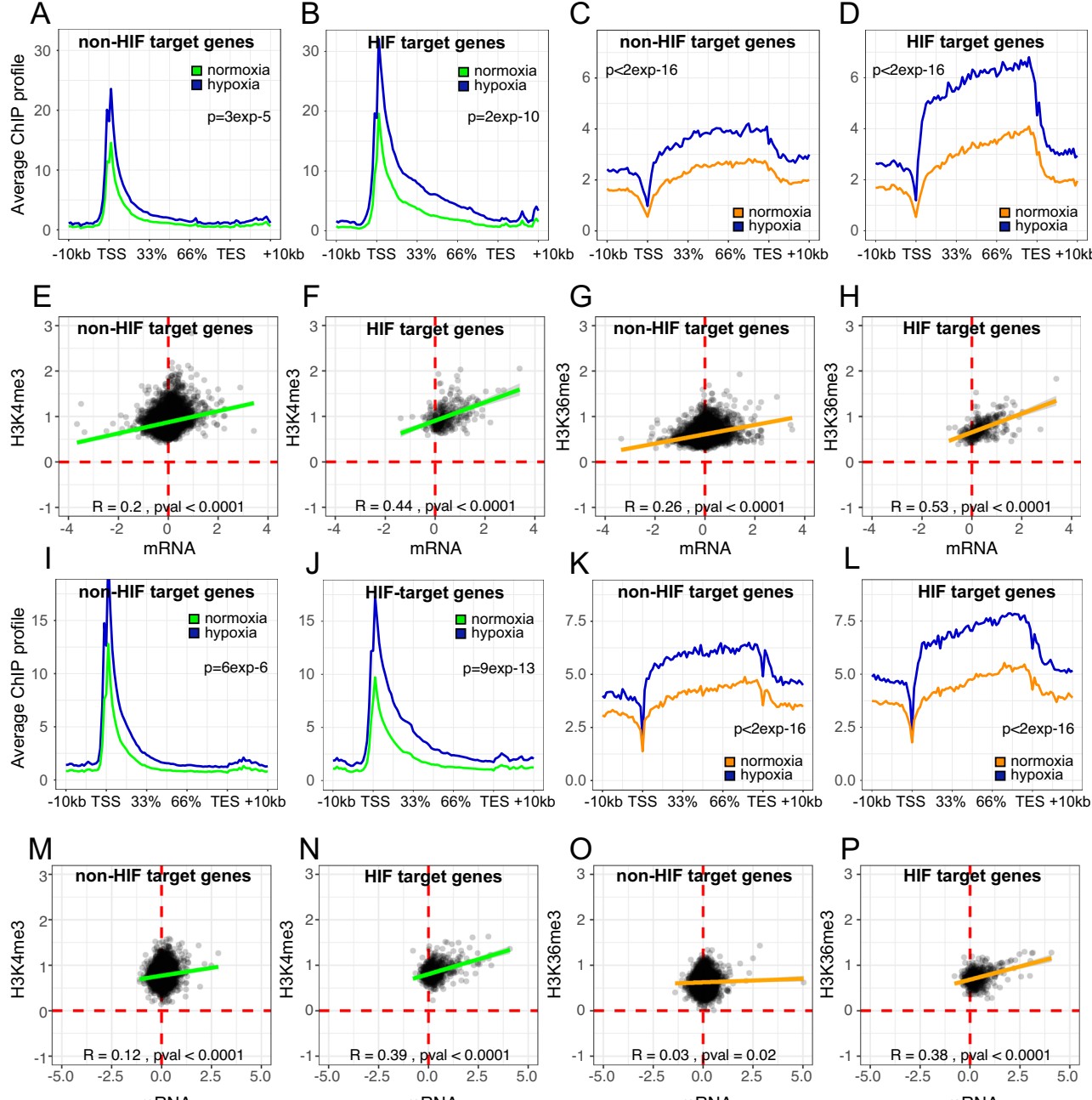

**Fig. 5 | Exaggerated hypoxic induction of H3K4me3 and H3K36me3 trimethylation at HIF target genes.** Average ChIP-seq signal for H3K4me3, normalised to Drosophila spike-in control, in PC3 cells ($n = 2$) incubated in 21% $O_2$ normoxia (green) or 16 h of 0.5% $O_2$ hypoxia (blue) across **A** a random set of non-HIF target genes or **B** HIF target genes. HIF-target genes were defined as the nearest TSS to a HIF binding site). Non-HIF targets more than 0.5 Mb away from an HIF binding site were randomly sampled and matched to the HIF target genes for both number ($n = 167$) and expression level. **C, D** The same plots showing H3K36me3 signal in 21% $O_2$ normoxia (orange) or 16 h of 0.5% $O_2$ hypoxia (blue). Scatter plots showing log2 fold-change in transcript abundance against log2 fold-change in H3K4me3 signal in hypoxia (0.5% $O_2$, 16 h) across the gene body (±2 kb) for **E** each non-HIF target gene and for **F** each HIF target gene in PC3 cells. **G, H** The same plots for H3K36me3 signal. **I–P** The same analyses in HCT116 cells. Pearson's correlation test was used to determine the correlation between hypoxic changes in histone methylation at each locus and changes in gene expression.

RNApol2, phospho-Ser5 RNApol2 and particularly phospho-Ser2 RNApol2 was observed at HIF target genes compared to non-HIF target genes. This correlates with enhanced H3K4me3 and H3K36me3 signals seen at these loci in hypoxia and is consistent with the hypothesis that both HIF and HIF-dependent activation of RNApol2 may lead to in cis changes in histone H3K4me3 and H3K36me3 superimposed upon global upregulation of these modifications in hypoxia. To determine causality and the dependence of these changes at HIF target gene loci on HIF, we examined the effect of hypoxia on histone H3K4me3 and H3K36me3 trimethylation in PC3 cells

in which the HIF transcriptional response had been blocked by CRISPR-Cas9 targeted deletion of the common HIF-1β subunit.

**HIF-1β knockout abrogates the exaggerated hypoxic induction of histone H3K4me3 and H3K36me3 trimethylation at HIF target genes, whilst preserving their global induction**

Two independent PC3 HIF-1β knockout clones were generated using different guide RNAs directed against exon 6 of the gene. Effective *HIF-1β* gene

**Fig. 6 | Hypoxic activation of RNApol2 at HIF target genes in hypoxia.** Average ChIP-seq signal for total RNApol (NTD), normalised to Drosophila spike-in control, in PC3 cells ($n = 2$) incubated in 21% $O_2$ normoxia (purple) or 16 h of 0.5% $O_2$ hypoxia (blue) across **A** a random set of non-HIF target genes or **B** HIF target genes as defined in Fig. 5. **C**, **D** The same plots showing phosphor-Ser5 RNApol2 signal in in 21% $O_2$ normoxia (green) or 16 h of 0.5% $O_2$ hypoxia (blue) and (**E**, **F**) phosphor-Ser2 RNApol2 signal in in 21% $O_2$ normoxia (yellow) or 16 h of 0.5% $O_2$ hypoxia (blue).

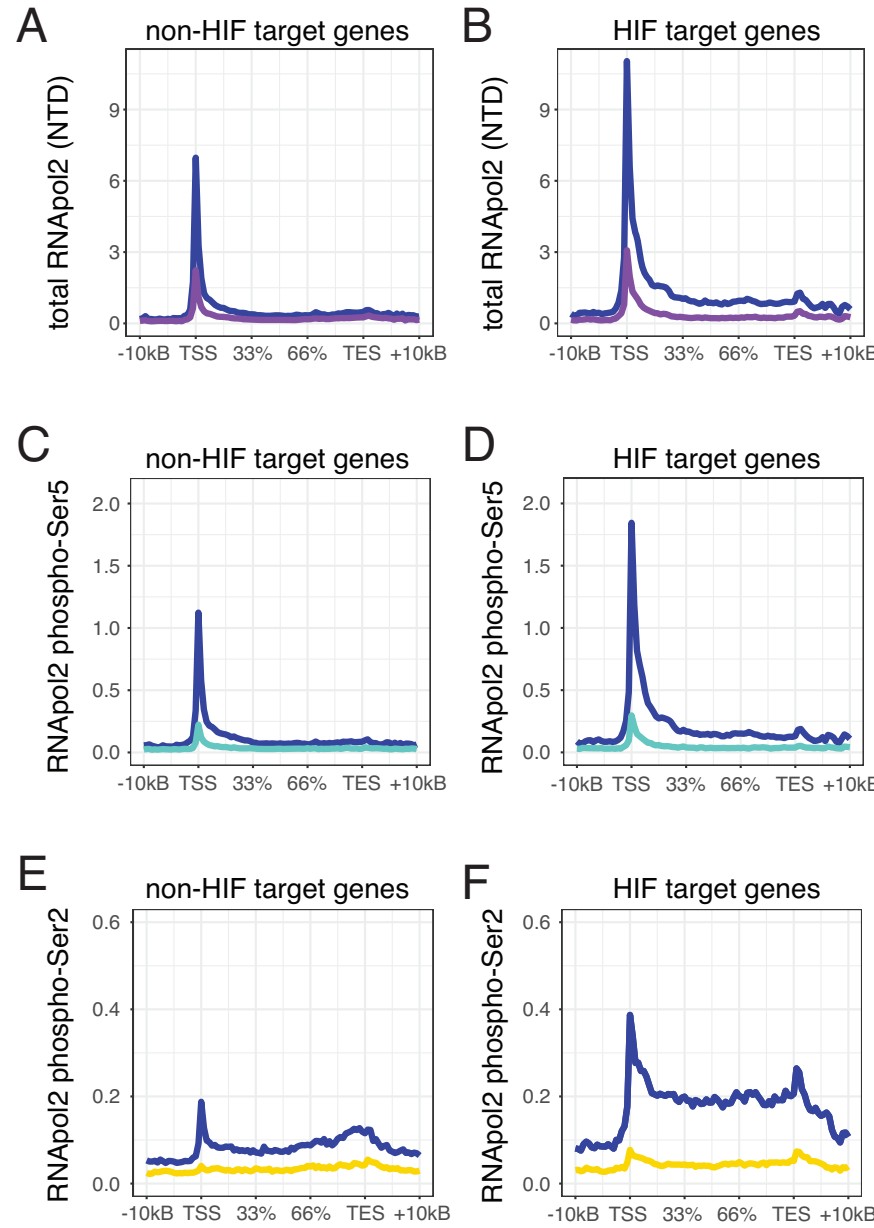

knockout was demonstrated by immunoblot for HIF-1β and for the HIF target genes n-myc downregulated gene 1 (NDRG1) and 6-phosphofructo-2-kinase/fructose-2,6-biphosphatase 4 (PFKFB4) (Fig. 7A). ChIP-qPCR analysis of HIF-1β and HIF-1α binding and qPCR analysis of additional HIF target genes further confirmed effective ablation of the HIF response (Supplementary Fig. 7). RNA-seq analysis of these cells also revealed widespread depletion of the transcriptional response to hypoxia, although significant regulation of a small number of genes was still observed (Fig. 7B, C).

Bulk changes in histone trimethylation were then determined by ChIP-Qubit analysis (Fig. 7D–G). Hypoxic induction of all four histone tri-methylation marks was comparable in HIF-1β KO cells and HIF-1β WT cells. Drosophila-normalised ChIP-seq analysis showed similar results, reinforcing these findings (Fig. 7H–K). Taken together, this indicates that the HIF transcriptional response does not greatly alter the global induction in histone trimethylation seen in hypoxia.

We then examined the effect of HIF-1β KO on the exaggerated induction of histone H3K4me3 and H3K36me3 trimethylation observed in hypoxia at HIF target genes. While no effect was observed on the induction

of either mark at non-HIF target genes, loss of HIF-1β abrogated the exaggerated hypoxic response in H3K4me3 and H3K36me3 observed at HIF target genes (Fig. 7L–S). Similar results were also observed when individual HIF-target gene loci were examined (Supplementary Fig. 8). Taken together, these results indicate a generalised induction of H3K4me3, H3K36me3, H3K9me3 and H3K27me3 trimethylation across the genome that is independent of the direction of hypoxic gene regulation together with a superadded HIF-dependent induction in H3K4me3 and H3K36me3 at HIF target genes. While the former is independent of HIF and transcriptional activation, the latter is dependent on both, indicating that it may arise either as a result of HIF itself or through HIF-mediated transcriptional activation.

## Discussion

Consistent with previous reports, we show that hypoxia leads to changes in total cellular histone trimethylation that can be detected in bulk analyses. Therefore, it is not surprising that we also observe that these changes are widespread across the genome in pangenomic analyses. Indeed, all 4 tri-methylation marks studied increase in hypoxia across the overwhelming

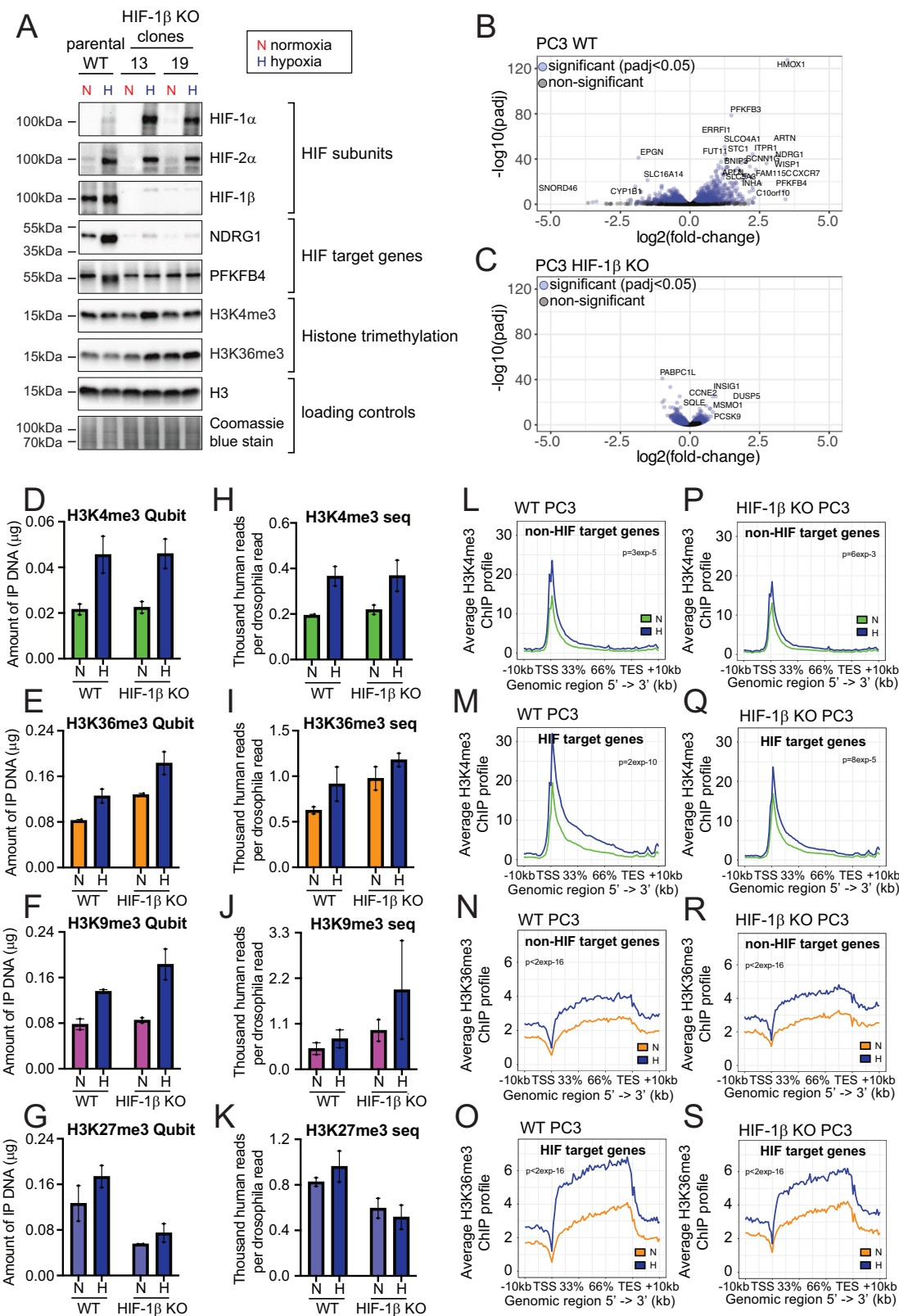

majority of gene loci (Fig. 3I–P). Furthermore, the widespread extent of these changes had been underappreciated by previous pangenomic studies in which normalisation methods had not accounted for global changes in histone trimethylation. This emphasises the importance of our methodological advance using spike-in normalisation methods when performing

pangenomic analyses of widespread modifications whose total abundance is altered between conditions.

Furthermore, these widespread increases in histone methylation occur in the absence of changes in total transcript abundance at this time point. This is similar to the findings of Perez et al.[32] who showed that high

**Fig. 7 | HIF-1β KO specifically abrogates hypoxic H3K4me3 and H3K36me3 induction at HIF-target genes but not global increases in histone trimethylation.** **A** Representative immunoblots showing the indicated protein levels in extracts from parental wild-type (WT) PC3 cells, and two independent clones targeted for CRISPR Cas9 HIF-1β deletion (clones #13 and #19), incubated in 21% O₂ normoxia (N, red) or 16 h of 0.5% O₂ hypoxia (H, blue). Volcano plots showing -log10(adjusted p-value) against log2(fold-change) for RNA-seq analysis of **B** PC3 and **C** PC3 HIF-1β knock-out (KO) cells (clone #13 and #19) incubated in normoxia (21% O₂) or hypoxia (0.5% O₂, 16 h). Each sample was normalised to ERCC spike-in control (n = 4 replicates). Qubit quantification of DNA immunoprecipitated from parental wild-type (WT) and HIF-1β knock-out (KO) PC3 cells incubated in 21% O₂ normoxia (N) or 16 h of 0.5% O₂ hypoxia (H, blue) using antibodies directed against **D** H3K4me3, **E** H3K36me3, **F** H3K9me3, or **G** H3K27me3 (n = 2 independent ChIP experiments). Error bars represent standard deviation with individual data points superimposed. Total read count, normalised to Drosophila spike-in, for ChIP-seq analysis of **H** H3K4me3, **I** H3K36me3, **J** H3K9me3, and **K** H3K27me3 in parental wild-type (WT) PC3 and HIF-1β knock-out (KO) cells incubated in normoxia (N) or 16 h at 0.5% O₂ hypoxia (H, blue). n = 2 independent experiments. Error bars represent standard deviation. Average ChIP-seq signal for H3K4me3, normalised to Drosophila spike-in control, across a random set of **L** non-HIF target genes or **M** HIF target genes, in Parental wild-type (WT) PC3 (n = 2) incubated in 21% O₂ normoxia (green) or 16 h of 0.5% O₂ hypoxia (blue). **N, O** The same plots for H3K36me3T. **P–S** The same analyses in HIF-1β CRISPR-Cas9 KO PC3 cells (n = 2).

expression of PEMT (Phosphatidylethanolamine N-Methyltransferase), which can act as a S-adenosyl methionine (SAM) sink, a co-factor for histone methylation, was associated with globally low levels of H3K4me3 and H3K9me3, but that this did not correlate with gene expression. Similarly, Clouaire et al. showed that depletion of Cfp1 (CXXC finger protein 1), a key component of the SET1A/B complex responsible for the bulk of H3K4me3 in mouse embryonic stem cells, resulted in a drastic loss of H3K4me3 again with minimal consequences for transcription[33]. In addition, we observe hypoxic increases in all four trimethylation marks at both upregulated and downregulated genes. This might indicate a degree of dissociation between changes in histone methylation and gene transcription induced by hypoxia such that the direction of changes in histone methylation do not predict the direction of changes in gene transcription. This may be because hypoxia increases histone modifications that are associated both with gene activation and gene silencing, the effects of which may be balancing each other out. However, one caveat to this work is that it has only examined a single timepoint that has been optimised to study the HIF response, and another possibility is that changes in histone methylation may take longer to generate transcriptional changes than those induced by binding of HIF. Furthermore, there may be threshold effects, whereby small changes in histone methylation are insufficient to cross a functional threshold necessary to alter recruitment of chromatin readers and to impact on biological processes. Similarly, if the availability of histone readers were limited, global changes in histone modification may not cause a redistribution between loci. Alternately, many histone readers recognise specific combinations of histone modification, and it is combinatorial changes rather than individual marks that are important in modulating gene expression[34,35]. Under this model, global changes in histone trimethylation may only result in changes in gene expression at a subset of gene loci where additional, specific, permissive modifications already exist.

Our findings also raise questions as to whether these global increases in histone trimethylation may have roles beyond transcriptional regulation. Of note, Batie et al.[6] showed both rapid induction of histone trimethylation in hypoxia and rapid loss upon re-oxygenation. This indicates both rapid writing and erasing of these marks and therefore, in the steady state a high rate of turnover would exist. Given their abundance, this methylation of histones, consumes large quantities of SAM, the major methyl donor in the cell[36] and is, in turn, energetically demanding. Therefore, mechanisms that reduce this turnover (by reducing demethylation) may be advantageous to cells in hypoxia. Other established roles for histone methylation that may be altered by hypoxia include regulation of alternate RNA spicing, X-chromosome inactivation, cell differentiation, euchromatin formation and DNA repair[37–39]. Global changes in histone modification, specifically H3K4me3 and H3K36me3 play a role in regulating alternate RNA-splicing[37]. Notably, hypoxia also leads to altered RNA splicing, although the role of histone methylation in this process has not been studied[40]. Similarly, hypoxia can induce reversible changes in global chromatin compaction detectable by electron microscopy and resistance to digestion by DNAseI[29]. Several histone methylation marks, including H3K4me3, H3K9me3 and H3K36me3 have also been implicated in the DNA-damage response[38]. While the effects of hypoxia on DNA damage are unclear, it has been

suggested that re-oxygenation following hypoxia produces reactive oxygen species that damage DNA[41]. Therefore, hypoxic increases in histone methylation may prime cells to cope with the stress of re-oxygenation, although given the rapid disappearance of these marks any effect would likely only be transient.

Notwithstanding the above, we observe a degree of variation in the hypoxia-induced changes in histone trimethylation between gene loci, which correlates weakly with gene expression. Furthermore, hypoxic changes in H3K4me3 and H3K36me3 (both markers of gene activation) are particularly marked at gene loci bound and transcriptionally activated by the HIF transcription factor. While hypoxia-induced changes in DNA accessibility[21,42–45] may facilitate access to these loci to enable greater deposition of H3K4/36me3, this association is largely abrogated by genetic inactivation of HIF, suggesting that it is either a direct or indirect consequence of HIF-mediated transcriptional activation. Indeed, Ortmann et al.[46] have shown that HIF directly recruits the SET1b histone methyltransferase to gene loci leading to increased H3K4me3. In this respect, H3K4me3 helps recruit the NuRF chromatin re-modelling complex to increase chromatin accessibility[47] and Batie et al. have shown that changes in chromatin accessibility in hypoxia, which correlated with H3K4me3 levels, were dependent upon HIF[45]. Therefore, it seems likely that changes in H3K4me3 are driving changes in accessibility rather than vice versa. In addition, activated RNApol2 can recruit the SETD2 histone methyltransferase leading to increased H3K36me3[48]. Therefore, it is likely that other transcription factors induced in hypoxia (either directly or through their transcriptional regulation by HIF) may also contribute to changes in histone methylation in hypoxia. Consistent with these hypotheses, using time-resolved analyses, Le Martelot et al. have shown that in response to clock (a member of the same family of bHLH-PAS transcription factors as HIF) changes in histone H3K4me3 and H3K36me3 lagged behind changes in transcription suggesting that they can be consequent upon rather than driving transcriptional regulation[49]. However, while these hypoxic changes at HIF-target gene loci may be driven by HIF, they may also be a necessary part of HIF-mediated transcriptional activation and may serve to amplify the transcriptional response at these loci through heightened recruitment of reader proteins. It would therefore be interesting to determine whether these increases in H3K4me3 and H3K36me3 correlated with increased binding of these readers.

In summary, our data suggest two mechanisms driving hypoxia-inducible changes in histone trimethylation. The first is a global increase, which might be independent of the direction of change in gene transcription. The second occurs at HIF target genes in response to HIF, likely through the previously described recruitment of histone methyltransferase activity by HIF and/or by activated RNApol2.

## Methods

### Cell culture

A549, HCT116 and T47D cells were purchased from ECACC; HepG2 cells were purchased from ATCC; and PC3 cells were validated by STR genotyping. RCC4 cells were a gift from C.H. Buys. A549 cells were grown in Ham's F12K medium; HCT116 cells were grown in McCoy's 5a medium;

HepG2, RCC4, RCC4-VHL and T47D cells were grown in DMEM; and PC3 cells were grown in DMEM/Hams F12 medium. All cell lines were grown at 37 °C in a humidified incubator with 5% $CO_2$ with 100 U/mL penicillin, 100 mg/mL streptomycin, 2 mM L-glutamine and 10% foetal bovine serum (Sigma-Aldrich) and regularly tested for mycoplasma infection. Hypoxia incubations were performed at 0.5% $O_2$ for 16 h in a hypoxia chamber (In Vivo2 400 Hypoxia Work Station, Ruskinn Technology).

### Generation of CRISPR Cas9 HIF-1β knock-out (KO) cells

CRISPR (clustered regularly interspaced short palindromic repeats) Cas9-mediated knock-out of HIF-1β in PC3 cells was performed by the University of Oxford Genome Engineering Oxford (GEO) Facility (https://www.path.ox.ac.uk/research-group/genome-engineering-oxford-geo-facility/). Briefly, synthetic crRNAs (CRISPR RNAs) (Sigma) were annealed to tracrRNA (trans-activating CRISPR RNA) (Sigma) to make full-length gRNA (guide RNA). Two pairs of crRNA sequences were used to target exon 6 of *HIF-1β* employing a common forward guide (5′-TCCATC TTCTATAACCCTAA-3′) and one of two reverse guides (5′-GTGA AATTGAACGGCGGCGA-3′ or 5′-TCTCACATGAAGTCCTTGCG-3′). 1 μg gRNA was combined with 2.5 μg Cas9 protein (IDT) and nucleofections (200,000 PC3 cells) performed using an Invitrogen Neon Nucleofector with the following programme: 1350 V, 10 ms pulse width, three pulses. Targeted cells were then cloned under antibiotic selection and independent clones from each gRNA pairing used for subsequent experiments.

### Western blotting

Urea-SDS cell lysates were prepared in lysis buffer (6.7 M Urea, 10% glycerol, 1%SDS, 10 mM Tris/HCl pH 6.8) supplemented with cOmplete protease inhibitor cocktail (Roche), and 1 mM DTT. Lysates were sonicated using a Vibra-cell ultrasonic processor (Sigma, Z412619) set to 30% amplitude for 5 cycles of 10 s ON/10 s OFF. Fixed-chromatin cell lysates were prepared as for chromatin immunoprecipitation. Protein concentrations were determined by BCA assay (Pierce) and normalized. Samples were prepared for SDS-PAGE by adding Laemmli buffer and 0.1 M DTT and heated to 95 °C for 3 min. Proteins were separated using AnyKD Mini-PROTEAN TGX precast gels (Bio-Rad) and mini-PROTEAN tetra cells (Bio-Rad) and transferred to Immobilon-P polyvinylidene fluoride membranes using mini-trans-blot cells (Bio-Rad). All Western blotting solutions were prepared in PBST (PBS containing 0.1% tween). Membranes were blocked in either 5% BSA fraction V or 5% milk for 1 h, before being incubated in primary antibodies for 2 h at room temperature. Primary antibodies used were anti-HIF-1α (BD cat no. 610959, 1:1000), anti-HIF-2α (Cell Signaling cat no. 7096S, 1:200), anti-HIF-1β (Cell Signaling cat. no. 5537S, 1:1000), anti-NDRG1 (Cell Signaling cat no. 5196S, 1:1000), anti-CA9 (Abcam cat no. ab15086, 1:500), anti-PFKFB4 (Novus Biological cat no, NBP2-97694, 1:1000), anti-H3K4me3 (Cell Signaling, cat no. 9751S, 1:3000), anti-H3K9me3 (Cell Signaling, cat no. 13969S, 1:3000), anti-H3K27me3 (Cell Signaling, cat no. 9733S, 1:3000), anti-H3K36me3 (Cell Signaling, cat no. 4909S, 1:3000), anti-H3 (Abcam cat no. ab1791, 1:20,000), and anti-b-actin (Abcam cat no. ab49900, 1:25,000). HRP-conjugated secondary antibodies (Dako) were incubated for an hour. Membranes were developed using SuperSignal West Dura Extended Duration Substrate (Thermo Scientific) and imaged using a Chemidoc XRS+ imager (Bio-Rad).

### Chromatin immunoprecipitation

Cells were grown on 15 cm dishes to 80–90% at harvest. Protein-DNA complexes were crosslinked by adding formaldehyde to a concentration of 1% (w/v) and gently rocking on ice for 10 min. The reaction was quenched by addition of glycine to a concentration of 125 mM, followed by 10 min of gentle rocking at room temperature. Cells were washed twice with PBS and scraped into 5 ml PBS. Cells were centrifuged for 10 min at 4 °C, 800 rpm and resuspended in 500 μl of ChIP SDS lysis buffer (50 mM Tris pH 8.1, 10 mM EDTA, 1% SDS). After 10 min, samples were diluted 1:1 with ChIP dilution buffer (16.7 mM Tris pH 8.1, 167 mM NaCl, 1.2 mM EDTA, 0.01% SDS, 1.1% Triton X-100). Samples were sonicated in 15 mL Bioruptor Plus

TPX sonication tubes using a Diagenode Bioruptor Plus water bath sonicator with sonication cycles of 15 s on/15 s off for 30 min at high intensity. Samples were centrifuged for 10 min at 4 °C, 13,000 rpm and the supernatant was collected.

Aliquots of each sample were then quantified. Briefly, cross-linking was reversed overnight at 65 °C, with shaking at 1400 rpm using an Eppendorf ThermoMixer F1.5 (USA Scientific, 4053-8420). Proteins were then digested with 2 μl of Proteinase K (~20 mg/ml, Thermo Fisher, FEREO0491) at 45 °C, with shaking at 1400 rpm for 4 h followed by RNA degradation, 1 μl of RNaseA (10 mg/ml, Thermo Fisher, FEREN0531) at 37 °C, 1400 rpm for a further 30 min. Samples were then mixed with 1 ml PB binding buffer (Qiagen, 19066) and 10 μl 3 M sodium acetate pH 5.2, and DNA was purified with MinElute PCR Purification columns (Qiagen, 28006) and eluted into 20 μl of nuclease-free water (Thermo Fisher, AM9937. Chromatin fragmentation was then assessed using Agilent 2200 TapeStation System Automated Electrophoresis and D1000 DNA ScreenTape analysis (Agilent 5067-5586, 5067-5583, 5067-5602), with optimum sizes ~200 bp. The concentration of purified DNA was quantified by Qubit 2.0 fluorometer (Invitrogen) and dsDNA Broad Range Assay kit (Thermo Fisher, Q32850).

5 μg of chromatin from each sample was diluted with ChIP SDS lysis buffer to a final volume of 1 ml in 2.0 ml Eppendorf DNA LoBind safe-lock microcentrifuge tubes (Thermo Fisher, 13-698-792) and 15 μl immunoprecipitating antibodies were added. Antibodies used were anti-H3K4me3 (Cell Signaling, 9751S), anti-H3K9me3 (Cell Signaling, 13969S), anti-H3K27me3 (Cell Signaling, 9733S), anti-H3K36me3 (Cell Signaling, 4909S), anti-H3 (Abcam, ab1791), anti-HIF-1α (PM14), 15ul of anti-HIF-2α (PM9), 10ul of anti-HIF-1β (Novus Biologicals NB100-110), anti-Rbp1 NTD (Cell Signaling, 14958), anti-Phospho-Rbp1 CTD Ser2 (Cell Signaling, 13499), or anti-Phospho-Rbp1 CTD Ser5 (Cell Signaling, 13523). In addition, 10 ng of Drosophila melanogaster chromatin (Active Motif, 53083) and 1 μg of Drosophila-specific histone variant H2Av antibody (Active Motif, 61686) were added to each reaction as a spike-in control for downstream normalisation. Chromatin/antibody reactions were incubated overnight at 4 °C on a Cole-Parmer Stuart Tube Rotator (Thermo Fisher, 50-196-0426). Antibody-bound DNA was recovered using magnetic Protein G Dynabeads (Thermo Fisher, 10004D). 50 μl of beads were added to each sample along with 500 μl ChIP dilution buffer and incubated on an end-over-end rotator for 2 h at 4 °C. Supernatant was removed using a magnetic rack and the beads were washed once in low salt wash buffer (20 mM Tris pH 8.1, 150 mM NaCl, 2 mM EDTA, 0.1% SDS, 1% Triton X-100), once in high salt wash buffer (20 mM Tris pH 8.1, 500 mM NaCl, 2 mM EDTA, 0.1% SDS, 1% Triton X-100), once in LiCl wash buffer (10 mM Tris pH 8.1, 250 mM LiCl, 1 mM EDTA, 1% sodium deoxycholate, 1% Igepal), and twice in TE buffer (10 mM Tris pH 8.0, 1 mM EDTA). 120 μl of freshly made elution buffer (0.1 M $NaHCO_3$, 1%SDS) was added to each ChIP and incubated in a thermomixer at room temperature/1400 rpm for 15 min. Samples were centrifuged, the supernatant collected, and another 120 μl added to perform a second elution as above, to obtain 240 μl eluate per ChIP. Final immunoprecipitated (IP) material and input controls were de-crosslinked and purified as described above. Purity was determined using a NanoDrop 2000 with optimum 260/280 absorbance ratios of 1.8. Finally, IP′d DNA was quantified by Qubit Fluorometer 2.0, using the dsDNA High Sensitivity Assay kit (Thermo Fisher, Q32851).

Chromatin immunoprecipitations for HIF-1α, HIF-2α and HIF-1β were performed as described above, with the following modifications—no quantification of chromatin was performed and no Drosophila spike-in was added. Instead, each immunoprecipitation reaction contained the total harvested chromatin from an entire 15 cm plate of cells at ~80–90% confluency. Sonicated chromatin was diluted 1:6 with ChIP dilution buffer and 40 μl of pre-washed Protein A agarose beads were added to pre-clear each sample for 1 h on end-over-end rotator, 4 °C before immunoprecipitation. Alongside the addition of antibodies, 15 μl rabbit pre-immunisation serum was used as a negative control. Antibody-bound DNA was pulled down with 45 μl of pre-washed Protein A agarose beads by rotating at 4 °C for 1.5 h. Samples were centrifuged for 8 min at 380 *g*, 4 °C. Washing of the beads

proceeded as described above, except that centrifugation was used instead of a magnetic rack. All ChIP-seq experiments were performed in duplicate in accordance with ENCODE Consortium guidelines.

## RNA extraction

Cells were grown on 6 cm dishes to approximately 80–90% confluency on the day of harvest. Total RNA was prepared using the Qiagen RNeasy Plus Mini Kit (Qiagen, 74134), and RNase-Free DNase Set (Qiagen, 79254) according to manufacturer's instructions. 1–4 μl of 1:10 diluted ERCC RNA Spike-in Mix 1 (Ambion, 4456740) was added to each sample. RNA purity was determined by NanoDrop 2000 and its integrity was assessed by Agilent 2200 TapeStation System Automated Electrophoresis and RNA ScreenTape analysis (Agilent 5067-5576, 5067-5578, 5067-5577). RNA was quantified by Qubit Fluorometer 2.0, RNA High Sensitivity Assay kit (Thermo Fisher, Q32852).

## qPCR

qPCR was performed on a StepOnePlus Real-Time cycler (Applied Biosystems) using Fast SYBR Green master mix (Thermo Fisher, 4385612), according to manufacturer instructions. Complimentary DNA (cDNA) was synthesised from purified RNA with the High-Capacity cDNA Reverse Transcription Kit (Thermo Fisher, 4368814).

ChIP-qPCR primer sequences were CA9-up (F: 5′-CGGGGAG ACTCAGGAAACAA-3′, R: 5′-TGTGGACTCTGACCCTGGAT-3′), CA9-prom (F: 5′-TGCCCACAGGGACAAAGAAG-3′, R: 5′-GGGTA TGGGAGGGGAGTCAT-3′), EGLN3-up (F: 5′-GGTTGTAAAGCATG GGCAGT-3′, R: 5′-GCTGATCGGGAGGTCCTTGA-3′), EGLN3-prom (F: 5′-GACAGGGCACGATGTACTCC-3′, R: 5′-GGATCCCGGACCT CGATTCT-3′), NDRG1-up (F: 5′-GGCCCTTGCCTGTCTCTTT-3′, R: 5′-TTCAAGCTGCAAATCTGGGC-3′), NDRG1-prom (F: 5′-CCCTTTCT GCTCCGCGTC-3′, R: 5′-CTTTCCTTTCTCCCTCGCGT-3′), NDRG1-HRE (F: 5′-TCCCTCCCAATCTCTCTCTTCTT-3′, R: 5′-CACCA TCAGCACAGCAAACTAC-3′), EGLN3-HRE (F: 5′-CCCAGTCACC AGAGAAATGTT-3′, R: 5′-GCGTCTTGATGTCCTTATCCCT-3′), Drosophila chromatin (Active Motif positive control primer set (71037) amplifies 92 bp region (exon1) of the Porphobilinogen synthase gene on the Drosophila melanogaster chromosome 3 L).

RT-qPCR primer sequences for analysis of cDNA were ERCC spike-in sequence 113 (F: 5′-CGCGCGTACTAGGCAGATAA-3′, R: 5′-CTCCGATAGCGACTGGACAC-3′), ERCC spike-in sequence 130 (F: 5′-CAACATGGTTCAACCGCCTG-3′, R: 5′-CGCTTGGCAGGACCATTT TT-3′), NDRG1 (F: 5′-TAACGTGGAAGTGGTCCACAC-3′, R: 5′-ATTGGTCGCTCAATCTCCAG-3′), CYP1B1 (F: 5′-GAGAACGTACCG GCCACTATC-3′, R: 5′-CGACCTGATCCAATTCTGCCT-3′), HPRT1 (F: 5′-CTGAGGATTTGGAAAGGGTGT-3′, R: 5′-CATCTCGAGCAAGA CGTTCA-3′), BNIP3 (F: 5′-GCTCCCAGACACCACAAGAT-3′, R: 5′-GAGAGTAGCTGTGCGCTTC-3′), PFKFB4 (F: 5′-AATTATCCACTG GAGTTCGCC-3′, R: 5′-AGCTCCATGATGACAGGCTC-3′), EGLN1 (F: 5′- GCAGCATGGACGACCTGATA-3′, R: 5′-CCATTGCCCGGATAA-CAAGC-3′), DDIT4 (F: 5′-GAACTCCCACCCCAGATCG-3′, R: 5′-TGTT CATCCTCAGGGTCACT-3′).

## High-throughput sequencing

In accordance with ENCODE Consortium guidelines, all ChIP experiments were performed in duplicate[50]. All libraries were prepared and sequenced by the US NCI CCR Genomics Core. Single-end sequencing was performed using NextSeq 500 (75 bp) or NextSeq 2000 (100 bp) platforms according to Illumina protocols.

In accordance with ENCODE Consortium guidelines, all RNA-seq experiments were performed in three or four biological replicates[51]. Messenger RNA (mRNA) isolation and library preparation were performed by the US NCI CCR Genomics Core using the NEB Ultra II RNA-seq kit and PolyA (mRNA) selection. Paired-end sequencing was performed using NextSeq 500 (75 bp) or NextSeq 2000 (100 bp) platforms according to Illumina protocols.

## Quantification and statistical analysis
### ChIP-seq
Alignment and normalisation. ChIP-seq reads were sequentially aligned to the Drosophila melanogaster BDGP6 genome and to the Human GRCh37 genome using BWA (0.7.5a-r405). The output SAM files were converted to BAM using SAMtools view (0.1.19) and the number of mapped reads was counted with SAMtools flagstat for later normalisation[52]. Reads mapping to both genomes and human reads mapping to the Duke Encode blacklist regions were excluded using Bedtools (2.17.0)[53]. Less than 0.001% of all reads mapped to both species. The final normalised read counts for each ChIP-seq sample were calculated by dividing the total number of Human mapped reads by the total number of Drosophila mapped reads to provide a scaling factor for normalisation of each sample.

Counting reads with samtools. ChIP-seq signal was quantified across protein-coding gene bodies using the Refseq[54] genebody.protein_coding annotation downloaded from the ngs.plot database[55]. Gene bodies were extended ±2 kilobases to include relevant ChIP signal up- and downstream of each gene and SAMtools bedcov (0.1.19)[52] was used to count the number of ChIPs-seq reads across each region. Read counts were then normalised according to the total number of Drosophila reads previously counted in each sample as above.

Visualisation with ngsplot. Ngs.plot.r[55] was used to visualise ChIP-seq signal across gene bodies and flanking regions using the following parameters: -G hg19 (genome name), -L 10000 (10,000 bp flanking region size), -IN 1 (tag for large gene body interval), -GO none (no particular ranking applied along y-axis of a heatmap), -D refseq (gene database).

Defining HIF binding sites. HIF ChIP-sequencing was performed by Dr. Olivia Lombardi and previously published (GEO: GSE200203, GEO: GSE130989)[30,31]. ChIP-seq peaks were identified as described[30], by the overlap of both T-PIC (Tree shape Peak Identification for ChIP-seq)[56] and MACS (model-based analysis of ChIP-seq)[57] peak callers. In order to be considered, a peak had to be present in both biological replicates (overlapping by at least one base pair, measured by Bedtools 2.17.10[53]). Finally, HIF binding sites were specifically defined as genomic loci where a HIF-1β ChIP peak overlapped a HIF-α peak (either HIF-1α or HIF-2α).

Pre-ranked gene set enrichment analysis (GSEA). Normalised ChIP-seq signal was quantified across gene bodies ±2 kb, as above, and the fold-difference between normoxia and hypoxia conditions calculated. Genes were then ranked in descending order according to this fold-change and gene set enrichment analyses (GSEA) was performed in pre-ranked mode using a weighted enrichment score and 10,000 permutations[58].

### RNA-seq
Alignment, normalisation and differential expression analysis. RNA-seq reads were sequentially aligned to the ERCC sequence library (ERCC92.fa) and to the Human GRCh37 genome using HISAT2 (2.05)[59]. The output SAM files were converted to BAM using SAMtools view (0.1.19) and the number of ERCC mapped reads was counted with SAMtools flagstat for later normalisation[52]. ERCC reads were removed prior to mapping to the Human GRCh37 genome. ERCC-mapped reads were quality checked for overlap by aligning to the Human GRCh37 genome. Less than 0.0001% of all reads mapped to both ERCC and GRCh37, and these were discarded. The final normalised read counts for each RNA-seq sample were calculated by dividing the total number of Human mapped reads by the total number of ERCC mapped reads to provide a scaling factor for normalisation of each sample. HTSeq (0.5.4p3) set to 'intersection-strict' mode was utilised to count the total number of reads mapping within each Refseq-defined gene[60]. Finally, differential expression between normoxia and hypoxia conditions was quantified using DESeq2[61].

**Pre-ranked gene set enrichment analysis (GSEA).** The fold-change in gene expression between normoxia and hypoxia conditions was calculated by DESeq2[61]. Genes were then ranked in descending order according to their fold-change and gene set enrichment analyses (GSEA) were performed in pre-ranked mode using a weighted enrichment score and 10,000 permutations[58].

**BigWig generation and IGV visualisation.** To display ChIP-seq and RNA-seq tracks using the Integrative Genomics Viewer (IGV 2.8.2)[62], BAM files were firstly indexed with SAMtools (0.1.19)[52]. Next, Bedtools genomecov (2.17.0) was used to assess genome-wide sequencing coverage[53]. The bedGraph output files were then converted to BigWig using the bedGraphtoBigWig tool. Refseq hg19 (GRCh37) build was used for chromosomal coordinates and gene annotation.

### Statistics and reproducibility

All ChIP-seq experiments were performed as biological duplicates and all RNA-seq experiments were performed as three or four biological replicates in accordance with ENCODE Consortium guidelines. Statistical methods are outlined in individual sections.

### Data availability

ChIP-seq and RNA-seq data are available at Gene Expression Omnibus (GEO): GSE296192, GSE313247, GSE200203 (HIF-1α and HIF-2α ChIP-seq), and GSE130989 (HIF-1β ChIP-seq). Original Western blot images are available in Supplementary Fig. 9 and source data are available in Supplementary Data. This paper does not report original code. Any additional information required to reanalyse the data reported in this paper is available from the lead contact upon request.

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

## Acknowledgements

This study was supported by the National Institute for Health Research (DRM, OL, SH & VL; NIHR-RP-2016-06-004), King Abdulaziz University, Ministry of High Education for Saudi Arabia (DRM & PJR) the NIH-Oxford-Cambridge Scholars Program (JDK), the Ludwig Institute for Cancer Research (PJR), the Wellcome Trust (PJR; 106241/Z/14/Z) and the Francis Crick Institute (PJR), which receives its core funding from Cancer Research UK (FC001501, the UK Medical Research Council (FC001501), and the Wellcome Trust (FC001501) and the Intramural Research Program, National Institutes of Health, Center for Cancer Research, National Cancer Institute, Center for Cancer Research (CHC & WDF; ZIA BC 010453). The computational aspects of this research were supported by the Wellcome Trust Core Award Grant Number 203141/Z/16/Z and the NIHR Oxford BRC. We thank Chris Pugh for scientific discussions. The views expressed are those of the author(s) and not necessarily those of the NHS, the NIHR or the Department of Health.

## Author contributions

Author contributions were as follows: conception and design of the work - JDK, CHC, WDF, PJR & DRM; acquisition, analysis, and interpretation of data – JDK, OL, SH, VNL & DRM, drafting and revising the work – JDK & DRM.

## Competing interests

The authors declare the following competing interests: P. J. Ratcliffe reports grants from Ludwig Institute for Cancer Research and Francis Crick Institute during the conduct of the study and personal fees from Immunocore plc outside the submitted work. No disclosures were reported by the other authors.
