## [Transparent Peer Review file · Communications Biology]

Hypoxic regulation of chromatin and gene transcription

Corresponding Author: Professor David Mole

Version 0:

Reviewer comments:

Reviewer #1

(Remarks to the Author)

Kindrick et al investigate histone methylation changes in prolonged hypoxia (16h) in several cell lines using a quantitative ChIP method. They validate previous findings that hypoxia increases global histone methylation, independently of HIF, and suggest that HIF is required for histone methylation at its own target genes. They also suggest histone methylation is disconnected from transcription, since most genes have both repressive and active marks and RNA seq in hypoxia. Their novel finding is a methodological one, including the spike in for ChIP-seq analysis. Most of the findings corroborate other findings in the literature including that HIF is required to bring histone methyl transferases to HIF-dependent target genes (Ortmann, 2021), which the last author is part of.

Some of the statements are not supported by the data. Given that it is almost impossible to deplete total histone marks, it is not possible to state that they are not important for transcription. In addition, this study was conducted at similar times of analysis for both histone modification and RNA analysis, so whether histone modifications are required for HIF access to their targets, or other transcription factors is also possible. It is clear that HIF is required for maintenance of some histone marks at its own targets, as previously suggested. However, many more genes are altered in hypoxia that are not HIF targets as well. So, my suggestion is to tone down some of the strong statements and include the limitations of the approach and study. Regardless this is an interesting addition to the field and a good method improvement

Regarding the statistic, these look ok, despite some of the ChIP-PCR analysis only using N=2.

Reviewer #2

(Remarks to the Author)

Comments to Authors

The manuscript by Kenrick et al. examines chromatin regulation and gene expression in hypoxia with a focus on histone methylation. Using ChIP-seq with a DNA spike-in normalisation strategy alongside RNA-seq, the authors show that histone trimethylation is broadly increased genome-wide in hypoxia. In contrast to prevailing models, these methylation gains do not consistently track with transcriptional changes. Notably, HIF-dependent genes display the largest increases in H3K4me3 and H3K36me3, suggesting a super-added, locus-specific effect at HIF targets. The methodological innovation around spike-in normalisation is a strength. The study is well written, experiments are carefully controlled, and the work will interest researchers in hypoxia, epigenetics, and transcription.

Major comments:

Visual exemplars of genome-wide claims

Please complement the aggregate analyses with genome browser snapshots for representative HIF-target and non-HIF-target loci (both marks that increase strongly and loci with minimal change). This will help readers contextualise the global effects.

To reinforce the conclusion that methylation changes can be uncoupled from transcription, consider ChIP-qPCR (H3K4me3, H3K36me3, H3K9me3, H3K27me3) and RT-qPCR at a small panel of genes (up-, down- and non-regulated; HIF-target vs non-target).

For both ChIP-seq and RNA-seq, please include PCA (or MDS) plots demonstrating replicate clustering and condition separation.

You show exaggerated methylation at HIF-bound loci. Could this reflect increased chromatin accessibility in hypoxia, enabling greater H3K4me3/H3K36me3 deposition? If feasible, adding or citing ATAC-seq/DNase-seq under matched conditions would clarify whether accessibility changes explain part of the effect.

Since the main analyses are in two cancer cell lines, consider repeating a key subset of experiments (e.g., bulk ChIP-Qubit and targeted ChIP-qPCR) in a non-transformed cell model to address whether the global increase is a general hypoxic response rather than cancer-specific.

The Discussion raises the possibility that heightened methylation at specific loci could cross a threshold for chromatin reader binding and thereby amplify transcription. If maintained, please expand briefly and, if possible, indicate how this could be tested (e.g., enrichment of specific readers at HIF targets).

Minor comments:

Error bars: Where appropriate, please report standard deviation (SD) rather than standard error (SEM) or justify the use of SEM in the legend.

Figure legibility: Increase axis/label font sizes across figures to improve readability.

Reviewer #3

(Remarks to the Author)

This manuscript uses spike-in normalized ChIP-seq to investigate the histone trimethylation landscape in hypoxia. The authors report a widespread, global increase in four key histone trimethylation marks that they conclude is largely HIF-independent and dissociated from transcriptional changes. They contrast this with a superimposed, HIF-dependent enhancement of H3K4me3 and H3K36me3 at direct HIF target genes. While the methodological approach is valuable, the manuscript has significant issues in data interpretation, oversimplified conclusions, and a failure to address the novelty and complexity of its findings. The authors' central claims are frequently based on an overstatement of the evidence, and the study lacks the direct mechanistic data required to support its conclusions.

Major Concerns

1. Abstract: The phrase "not a primary driver" overstates dissociation because the data show weak positive correlations between H3K4me3/H3K36me3 changes and gene induction; It is recommended to revise it as "Global H3 trimethylation increases in hypoxia are widespread and not sufficient to predict transcriptional direction, whereas enhanced H3K4me3/H3K36me3 at direct HIF targets appears consequent to HIF binding and transcriptional engagement" and avoid categorical causal language in Abstract and Discussion.
2. The conceptual finding that hypoxia leads to a global increase in histone methylation due to the oxygen-dependent activity of histone demethylases is already an established concept in the field, as acknowledged by several citations in the manuscript's introduction. The primary novelty of this work is the pangenomic quantification of this known phenomenon. The manuscript should be rewritten to frame its contribution accurately as a methodological advance and quantitative confirmation, rather than the discovery of a new biological principle.
3. The rationale for cell line selection is unclear and appears inconsistent. The initial screen in Figure 1 shows that T47D cells had a significant hypoxic induction of both H3K4me3 and H3K36me3. In contrast, PC3 cells did not show a significant induction of H3K36me3 in this assay. The decision to proceed with PC3 while excluding T47D is not adequately justified and weakens the foundation of the subsequent experiments.
4. The authors demonstrate a statistically significant global increase in histone marks using bulk quantification (ChIP-Qubit) and total sequencing reads. However, the biological relevance of a small average fold-change distributed across thousands of gene loci is questionable. The manuscript fails to discuss whether this modest global increase is sufficient to cross a functional threshold for altering the binding of chromatin readers or impacting biological processes.
5. The authors should add normalization of each mark to total H3 ChIP at matched loci (or input-normalized occupancy) to control for global changes in nucleosome density/compaction known to occur in hypoxia, ensuring that observed global increases are not driven by differential chromatin packing alone.
6. The manuscript claims "almost identical numbers of genes either up- or downregulated in hypoxia". This statement is not fully supported by visual inspection of the volcano plot for HCT116 cells (Figure 4B), which shows a clear asymmetry toward upregulation, consistent with HIF's primary role as a transcriptional activator. The authors must provide the exact numbers of significantly regulated genes to justify this claim, as it is critical to their central argument of dissociation.

7. This study will benefit from kinetic reversibility experiment. A brief re-oxygenation time course (e.g., 1–4 h) would demonstrate rapid reversibility of the global marks and clarify whether the HIF-dependent increment at targets shows distinct kinetics, aligning interpretation with prior observations of rapid writing/erasing under oxygen shifts.

8. A major weakness of this study is the absence of direct mechanistic experiments to support its claims. The link between HIF binding, transcriptional activation, and enhanced H3K4me3/H3K36me3 is merely correlational. To establish a causal link, the authors should provide additional data at representative HIF target genes.

Recommended experiments include:

- ChIP-qPCR or ChIP-seq for Pol II Ser2/Ser5 phosphorylation to directly measure transcriptional elongation.
- ChIP-qPCR or ChIP-seq for the relevant histone methyltransferases (e.g., SET1B, SETD2) to show their recruitment to HIF target genes in a hypoxia- and HIF-dependent manner.
- Perturbation experiments (e.g., siRNA/inhibitor) targeting these methyltransferases to demonstrate their necessity for the enhanced H3K4me3/H3K36me3 marks and gene expression.

Version 1:

Reviewer comments:

Reviewer #1

(Remarks to the Author)

The authors have addressed the vast majority of my concerns and have removed some of the over interpretation of their results. In addition, more data as been added which is of high quality. Some minor points still exist regarding over claiming.

Page 10, row 284, please add "at this time point" to the sentence after "transcript abundance".

Page 10 row 292, include the word "might" such as "This might indicate a..."

Page 12, row 359. please add "might be independent of the direction..."

Reviewer #2

(Remarks to the Author)

The authors have done an excellent job in answering all my queries and that of the other reviewers. They have provided additional experimental data that strengthens their conclusions and improves the manuscript overall.

Reviewer #3

(Remarks to the Author)

The authors have addressed all my concerns.

Reviewers' comments:

Reviewer #1 (Remarks to the Author):

Kindrick et al investigate histone methylation changes in prolonged hypoxia (16h) in several cell lines using a quantitative ChIP method. They validate previous findings that hypoxia increases global histone methylation, independently of HIF, and suggest that HIF is required for histone methylation at its own target genes. They also suggest histone methylation is disconnected from transcription, since most genes have both repressive and active marks and RNA seq in hypoxia.

Their novel finding is a methodological one, including the spike in for ChIP-seq analysis. Most of the findings corroborate other findings in the literature including that HIF is required to bring histone methyl transferases to HIF-dependent target genes (Ortmann, 2021), which the last author is part of. Some of the statements are not supported by the data. Given that it is almost impossible to deplete total histone marks, it is not possible to state that they are not important for transcription. In addition, this study was conducted at similar times of analysis for both histone modification and RNA analysis, so whether histone modifications are required for HIF access to their targets, or other transcription factors is also possible. It is clear that HIF is required for maintenance of some histone marks at its own targets, as previously suggested. However, many more genes are altered in hypoxia that are not HIF targets as well. So, my suggestion is to tone down some of the strong statements and include the limitations of the approach and study. Regardless this is an interesting addition to the field and a good method improvement

We acknowledge the limitations of our work and have revised our manuscript in line with the reviewer's comments above.

Regarding the statistic, these look ok, despite some of the ChIP-PCR analysis only using N=2.

Reviewer #2 (Remarks to the Author):

Comments to Authors

The manuscript by Kenrick et al. examines chromatin regulation and gene expression in hypoxia with a focus on histone methylation. Using ChIP-seq with a DNA spike-in normalisation strategy alongside RNA-seq, the authors show that histone trimethylation is broadly increased genome-wide in hypoxia. In contrast to prevailing models, these methylation gains do not consistently track with transcriptional changes. Notably, HIF-dependent genes display the largest increases in H3K4me3 and H3K36me3, suggesting a super-added, locus-specific effect at HIF targets. The methodological innovation around spike-in normalisation is a strength. The study is well written, experiments are carefully controlled, and the work will interest researchers in hypoxia, epigenetics, and transcription.

Major comments:

Visual exemplars of genome-wide claims

Please complement the aggregate analyses with genome browser snapshots for representative HIF-target and non-HIF-target loci (both marks that increase strongly and loci with minimal change). This will help readers contextualise the global effects.

We have added further supplemental figures showing IGV genome browser tracks of RNA-seq and ChIP-seq analysis of gene transcription, histone trimethylation and HIF binding to illustrate specific gene loci that show (Supplemental Figure 5) up/downregulation in hypoxia in each cell line, (Supplemental Figure 6) HIF-binding and HIF-dependent transcription in each cell line, and (Supplemental Figure 8) the effects of blocking HIF-dependent transcription in PC3 cells using HIF-1b CRISPR KO.

To reinforce the conclusion that methylation changes can be uncoupled from transcription, consider ChIP-qPCR (H3K4me3, H3K36me3, H3K9me3, H3K27me3) and RT-qPCR at a small panel of genes (up-, down- and non-regulated; HIF-target vs non-target).

Rather than performing ChIP-qPCR, which can only show signal at specific sites within a gene, we have added IGV genome browser tracks as above showing the signal across representative genes and flanking regions.

For both ChIP-seq and RNA-seq, please include PCA (or MDS) plots demonstrating replicate clustering and condition separation.

These have now been included as a new Supplemental Figure 2.

You show exaggerated methylation at HIF-bound loci. Could this reflect increased chromatin accessibility in hypoxia, enabling greater H3K4me3/H3K36me3 deposition? If feasible, adding or citing ATAC-seq/DNase-seq under matched conditions would clarify whether accessibility changes explain part of the effect.

This is an interesting suggestion. Any increase in accessibility would potentially allow greater access of both H3K4me3/H3K36me3 writers and erasers, so the overall effect of any changes in accessibility would be difficult to predict. Furthermore, HIF often binds many (tens) of kilobases away from its target gene promoter with which it interacts through chromatin looping, so any direct effect of HIF on displacing histones at its binding site may not alter accessibility at the transcriptional start site. In addition, HIF-target genes are typically already expressed in normoxia, with HIF binding merely enhancing the release of pre-bound promotor-paused RNAPol2, and therefore are already accessible to the transcriptional machinery (and potentially histone methylases)[Choudhry EMBO Rep 2014, Dengler Crit Rev Biochem Mol Biol 2014]. Nevertheless, as the reviewer points out, several groups have described increases in DNA accessibility in hypoxia at HIF target genes [Xin Nature Comm 2020, Li Cell Death Dis 2020, Wang IUBMB Life 2020, Ward eLife 2021 & Batie Biochem J 2022]. However, H3K4me3 can recruit the NuRF complex, which uses ATP to re-model the nucleosomes to open up chromatin structure and it is generally felt that changes in H3K4me3 are responsible for changes in accessibility rather than vice versa [Wysocka Nature 2006]. We have added a brief discussion and relevant citations to the manuscript (lines 334-343, page 12).

Since the main analyses are in two cancer cell lines, consider repeating a key subset of experiments (e.g., bulk ChIP-Qubit and targeted ChIP-qPCR) in a non-transformed cell model to address whether the global increase is a general hypoxic response rather than cancer-specific.

Consistent with the field in general, we have performed our pangenomic experiments in standard transformed laboratory cell lines for which protocols have been optimised. In this respect, the two cell lines that we use have distinct genetic and epigenetic contexts relevant to this purpose. In reports where both transformed and non-transformed cell lines have been used to study global hypoxic increases in histone methylation concordant results have been observed. Therefore, we felt that further confirmation of these findings in the context of the current study was not necessary. Furthermore, repeating the analysis hypoxic changes in histone trimethylation in relation to gene regulation and/or HIF binding in a non-transformed cell lines would entail considerable additional work

and is not feasible with available resources within the revision window. In particular, the hypoxic transcriptional responses differ from cell type to cell type, and so this work would require further pangenomic analysis of HIF-binding and transcriptional regulation to delineate HIF-target genes in a new cell line in addition to further ChIP analysis.

The Discussion raises the possibility that heightened methylation at specific loci could cross a threshold for chromatin reader binding and thereby amplify transcription. If maintained, please expand briefly and, if possible, indicate how this could be tested (e.g., enrichment of specific readers at HIF targets).

This is an interesting point. Whether or not there is a threshold effect, heightened recruitment of reader proteins by increased levels of H3K4me3 and H3K36me3 may be amplifying the response at specific loci. We have expanded upon this in the discussion (lines 299-301, page 11).

Minor comments:

Error bars: Where appropriate, please report standard deviation (SD) rather than standard error (SEM) or justify the use of SEM in the legend.

Apologies, there was a mistake in the legend for figure 1. The error bars were showing standard deviation rather than standard error of the mean as reported. We have corrected this oversight and have also added the individual data points to the graphs to show the full range of values for clarity.

Figure legibility: Increase axis/label font sizes across figures to improve readability.

We have increased the font sizes across all the main and supplemental figures as requested. The total file size and therefore resolution is limited by the submission process. However, we are also able to provide high resolution images of all figures for publication.

Reviewer #3 (Remarks to the Author):

This manuscript uses spike-in normalized ChIP-seq to investigate the histone trimethylation landscape in hypoxia. The authors report a widespread, global increase in four key histone trimethylation marks that they conclude is largely HIF-independent and dissociated from transcriptional changes. They contrast this with a superimposed, HIF-dependent enhancement of H3K4me3 and H3K36me3 at direct HIF target genes. While the methodological approach is valuable, the manuscript has significant issues in data interpretation, oversimplified conclusions, and a failure to address the novelty and complexity of its findings. The authors' central claims are frequently based on an overstatement of the evidence, and the study lacks the direct mechanistic data required to support its conclusions.

Major Concerns

1. Abstract: The phrase “not a primary driver” overstates dissociation because the data show weak positive correlations between H3K4me3/H3K36me3 changes and gene induction; It is recommend to revise it as “Global H3 trimethylation increases in hypoxia are widespread and not sufficient to predict transcriptional direction, whereas enhanced H3K4me3/H3K36me3 at direct HIF targets appears consequent to HIF binding and transcriptional engagement” and avoid categorical causal language in Abstract and Discussion.

We apologise for our choice of language and have amended the manuscript to reflect our findings more accurately, in line with the reviewer’s suggestions.

2. The conceptual finding that hypoxia leads to a global increase in histone methylation due to the oxygen-dependent activity of histone demethylases is already an established concept in the field, as acknowledged by several citations in the manuscript's introduction. The primary novelty of this work is the pangenomic quantification of this known phenomenon. The manuscript should be rewritten to frame its contribution accurately as a methodological advance and quantitative confirmation, rather than the discovery of a new biological principle.

The reviewer is correct in their assessment that our manuscript provides a methodological advance over previous papers in this field. Importantly, this has allowed us to properly normalise locus-specific

changes in histone methylation so that they can be correlated with changes in gene transcription. Conceptually, this has revealed a degree of dissociation between widespread changes in histone methylation and changes in gene transcription across most gene loci (i.e. increases in histone trimethylation are observed both at genes upregulated in hypoxia and at genes downregulated in hypoxia). Notwithstanding this finding, we do observe a correlation between hypoxia-induced H3K4me3 and H3K36me3 signal and hypoxic induction of gene transcription at HIF-target genes that is dependent on HIF activity. We have amended the manuscript to make these major findings clearer.

3. The rationale for cell line selection is unclear and appears inconsistent. The initial screen in Figure 1 shows that T47D cells had a significant hypoxic induction of both H3K4me3 and H3K36me3. In contrast, PC3 cells did not show a significant induction of H3K36me3 in this assay. The decision to proceed with PC3 while excluding T47D is not adequately justified and weakens the foundation of the subsequent experiments.

We acknowledge this apparent limitation. We did not want to choose the two most extreme cell lines as we felt that this may not adequately represent the typical situation. In addition, the PC3 cells line had the highest levels of HIF, facilitating the examination of the effects of HIF on histone trimethylation, whilst T47D cells had much lower levels of HIF.

4. The authors demonstrate a statistically significant global increase in histone marks using bulk quantification (ChIP-Qubit) and total sequencing reads. However, the biological relevance of a small average fold-change distributed across thousands of gene loci is questionable. The manuscript fails to discuss whether this modest global increase is sufficient to cross a functional threshold for altering the binding of chromatin readers or impacting biological processes.

This is an interesting point and one that we have now raised in the discussion (lines 299-301, page 11).

5. The authors should add normalization of each mark to total H3 ChIP at matched loci (or input-normalized occupancy) to control for global changes in nucleosome density/compaction known to occur in hypoxia, ensuring that observed global increases are not driven by differential chromatin packing alone.

In the current study, we have examined H3K4/9/27/26me3 signal normalised to the number of cells (i.e. to amount of chromatin) using a Drosophila spike in since we felt that this was more likely to correlate with gene expression than the amount of H3K4/9/27/26me3 per histone. Indeed, previous studies showing a correlation between basal histone methylation and transcription have not normalised to total H3 signal. However, the reviewer is correct that changes in nucleosome density/compaction are known to occur in hypoxia and may at least in part underly the hypoxic changes that we observe. Therefore, we have performed additional ChIP-seq analysis of total H3 methylation on the same chromatin we analysed for H3K4/9/27/26me3 in PC3 cells. This shows a slight increase in total H3 occupancy consistent with previously reported findings. When normalised to total H3 signal, increases in H3K4me3, H3K36me3 and H3K9me3 signal were still observed in hypoxia. Conversely, global changes in H3K27me3, which have previously been associated with chromatin compaction were unaltered by hypoxia when normalised to total histone signal. We have added these analyses as a new Supplemental Figure 3 and have discussed their implications in the text (lines 156-162, page 6).

6. The manuscript claims "almost identical numbers of genes either up- or downregulated in hypoxia". This statement is not fully supported by visual inspection of the volcano plot for HCT116 cells (Figure 4B), which shows a clear asymmetry toward upregulation, consistent with HIF's primary role as a transcriptional activator. The authors must provide the exact numbers of significantly regulated genes to justify this claim, as it is critical to their central argument of dissociation.

We apologise that the volcano plots appear misleading on this point. Figures 4C and 4D showing the log₂ fold change plotted against rank show this point more clearly and contrast with similar plots for the histone modifications (3I to 3P). This is confirmed by the absolute numbers of up/downregulated genes in each cell line (7,507 up vs 7,025 down for PC3 and 6,793 up vs 7,218 down for HCT116 cells) and by the number of up/downregulated genes reaching statistical significance in each cell line (1,416 up vs 1,250 down for PC3 and 2,007 up vs 1,852 down for HCT116 cells, adjusted p-value < 0.05). We have now included the total number of up/downregulated genes in the manuscript as we had already done for the histone marks to help make this point clearer (lines 187-188, page 7).

7. This study will benefit from kinetic reversibility experiment. A brief re-oxygenation time course (e.g., 1–4 h) would demonstrate rapid reversibility of the global marks and clarify whether the HIF-dependent increment at targets shows distinct kinetics, aligning interpretation with prior observations of rapid writing/erasing under oxygen shifts.

We have performed Western blot analysis of bulk histone H3K4/9/27/36me3 methylation after re-oxygenation and observed a reduction by 1 hour consistent with previous reports of rapid writing/erasing under oxygen shifts. However, this signal is likely to be dominated by global changes occurring at non-HIF target genes. Distinguishing time-dependent changes at HIF- vs non-HIF-target genes would require analysis of multiple further ChIP-seq experiments, which is beyond the scope of the current manuscript. Furthermore, the demonstration of distinct kinetics at HIF-dependent genes would be open to multiple interpretations and would not inform majorly on the mechanisms of writing/erasing at these loci.

8. A major weakness of this study is the absence of direct mechanistic experiments to support its claims. The link between HIF binding, transcriptional activation, and enhanced H3K4me3/H3K36me3 is merely correlational. To establish a causal link, the authors should provide additional data at representative HIF target genes.

Recommended experiments include:

- ChIP-qPCR or ChIP-seq for Pol II Ser2/Ser5 phosphorylation to directly measure transcriptional elongation.

As suggested, we have performed additional ChIP-seq experiments to examine total RNApol2 (NTD) and phosphor-Ser2/5 RNApol2, normalised to drosophila spike-in. These show enhanced, hypoxic induction of total RNApol2, phospho-Ser5 RNApol2 and particularly phospho-Ser2 RNApol2 at HIF target genes compared to non-HIF target genes. This correlates with enhanced H3K4me3 and H3K36me3 signals seen at these loci in hypoxia and is consistent with our hypothesis that phosphorylated RNApol2 is, at least in part, responsible for the enhanced trimethylation at these loci, through recruitment of histone methyltransferase activity. We have added these analyses as a new Figure 6 and discussed the results in the manuscript (lines 223-230, page 8).

- ChIP-qPCR or ChIP-seq for the relevant histone methyltransferases (e.g., SET1B, SETD2) to show their recruitment to HIF target genes in a hypoxia- and HIF-dependent manner.

ChIP-seq of a transiently bound enzyme is difficult and doesn't correlate well with activity (e.g. very active enzymes may only have a very short residency time). Furthermore, these experiments might be confounded by reduced levels of unmethylated substrate at sites where the trimethylation product is at its highest.

- Perturbation experiments (e.g., siRNA/inhibitor) targeting these methyltransferases to demonstrate their necessity for the enhanced H3K4me3/H3K36me3 marks and gene expression.

The main focus of our perturbation experiments was on the HIF transcription pathway (HIF-1beta CRISPR KO), since we felt that this was the most novel finding. Furthermore, while completely blocking methyltransferase activity might block hypoxic induction of H3K4me3/HH3K36me3 due to recruitment of these methyltransferases, the same would also be true if hypoxia were inhibiting the erasure of these marks, since any increase, would by necessity require the writing of new marks. Thus, siRNA and/or inhibition of these methyltransferases would not distinguish between these mechanisms.

REVIEWERS' COMMENTS:

Reviewer #1 (Remarks to the Author):

The authors have addressed the vast majority of my concerns and have removed some of the over interpretation of their results. In addition, more data as been added which is of high quality.
Some minor points still exist regarding over claiming.

Page 10, row 284, please add "at this time point" to the sentence after "transcript abundance".

The text has been altered as suggested

Page 10 row 292, include the word "might" such as "This might indicate a..."

The text has been altered as suggested

Page 12, row 359. please add "might be independent of the direction..."

The text has been altered as suggested

Reviewer #2 (Remarks to the Author):

The authors have done an excellent job in answering all my queries and that of the other reviewers. They have provided additional experimental data that strengthens their conclusions and improves the manuscript overall.

No further comment

Reviewer #3 (Remarks to the Author):

The authors have addressed all my concerns.

No further comment